

# Temporal fusion transformer-based strategy for efficient multi-cloud content replication

Naganandhini S. and Shanthi D.

Computer Science and Engineering, PSNA College of Engineering and Technology, Dindigul, Tamilnadu, India

## ABSTRACT

In cloud computing, ensuring the high availability and reliability of data is dominant for efficient content delivery. Content replication across multiple clouds has emerged as a solution to achieve the above. However, managing optimal replication while considering dynamic changes in data popularity and cloud resource availability remains a formidable challenge. In order to address these challenges, this article employs TFT-based Dynamic Data Replication Strategy (TD2RS), leveraging the Temporal Fusion Transformer (TFT), a deep learning temporal forecasting model. This proposed system collects historical data on content popularity and resource availability from multiple cloud sources, which are then used as input to TFT. Then TFT is used to capture temporal patterns and forecasts future data demands. An intelligent replication is performed to optimize content replication across multiple cloud environments based on these forecasts. The framework's performance was validated through extensive experiments using synthetic time-series data simulating with varied cloud resource characteristics. Some of the findings include that the proposed TFT approach improves the availability of data by 20% when compared to traditional replication techniques and also cuts down the latency level by 15%. These outcomes indicate that the TFT-based replication strategy targets to improve content delivery efficiency in the dynamic cloud computing environment, thus providing effective solution to dynamically address the availability, reliability, and performance challenges.

## INTRODUCTION

The provision of on-demand computer resources *via* the internet is referred as cloud computing (*Pham et al., 2023*). A shared pool of programmable computer resources, such as servers, storage, databases, networking, software, and applications, is made available to users. Users can use these resources as services, paying for what they use on a subscription or pay-as-you-go basis, in place of owning and maintaining physical infrastructure (*Islam, Karunasekera & Buyya, 2022*). Data replication is a critical aspect of cloud computing, particularly in the context of ensuring data availability. It becomes increasingly important as businesses rely on the cloud for their data storage and processing needs

Corresponding author
Naganandhini S.,
nandhu.be2010@gmail.com

(*Dustdar, Pujol & Donta, 2023*). It involves the duplication of data across multiple cloud environments or geographic regions, providing a robust solution to address several pressing challenges. Data replication plays a pivotal role in addressing various challenges in the cloud computing landscape. One of the primary challenges is ensuring high availability and reliability of data. Downtime, whether due to hardware failures, network issues, or other disruptions, can be detrimental to businesses (*Shahidinejad, Ghobaei-Arani & Masdari, 2021*). By replicating data across multiple cloud environments, organizations can mitigate these risks. In the event of an outage or failure in one cloud provider's infrastructure, data remains accessible from another which ensures business continuity. Data replication also serves as a cornerstone of disaster recovery strategies. Natural disasters, cyberattacks, or human errors can lead to data loss or system outages. By generating redundancy through data replication, businesses can quickly resume operations by utilizing replicated data from cloud environments. In today's fiercely competitive and fast-paced corporate environment, redundancy is essential for preventing data loss and minimizing downtime. Data replication can lower latency and improve speed in addition to disaster recovery. Organizations can optimize data access times by putting copies of the data in strategically located geographic areas or closer to end users. This is especially important for real-time analytics and content delivery networks, two applications that demand low-latency data retrieval (*Huang et al., 2020*; *Milani & Navimipour, 2017*). While data replication offers numerous advantages, implementing an effective strategy is complex. It necessitates careful planning, synchronization mechanisms, cost considerations, and adherence to compliance requirements, particularly when dealing with sensitive data (*Dabas & Aggarwal, 2019*).

Several recent approaches in content replication techniques for dynamic cloud environment are designed to enhance data accessibility, resource efficiency, response time, and flexibility in the changeable cloud environment. Static replication strategies are not effective for managing the complexity of cloud networks since the latter is dynamic. For this reason, machine learning and predictive models are currently being employed to make real-time replication decisions. Long short-term memory (LSTM) as well as Recurrent Neural Network (RNN) are among the models currently in use to predict future data usage and cloud resource variability (*Lim et al., 2021*). This allows replication to be more efficient, in terms of sparing resources whilst maintaining high availability. One of these is data popularity prediction models whereby an organization can predict the content that will be most popular. These models enable systems to replicate content proactively and that is due to the use of deep learning and TSF (time-series forecasting). This minimizes the time taken and keeps content ready for use to meet resource demand. Multi-cloud and edges computing have also enhanced the replication of strategies as a result of integrating the two. Replication is therefore spread across several cloud providers as well as edge resources to improve availability and minimize latency. Centralized cloud storage coupled with edge computing or a combination of both is useful in meeting user needs more comprehensively, especially in environments where such needs are limited by available resources. However, resource aware replication has been developed as a means of improving resource utilization in the cloud (*Milani & Navimipour, 2017*). In replication

decisions, availability of compute power, storage and bandwidths in real time is taken into consideration with reference to cost and performance. This has helped in improving the output of content delivery and costs been well managed.

The use of the Temporal Fusion Transformer (TFT) for content replication in cloud environments represents an innovative approach to deal with the challenges of data availability, reduction in latency, and efficient resource utilization (*Lim et al., 2021*). This cutting-edge technique uses the capabilities of TFT, a deep learning-based temporal forecasting model, to optimize content replication strategies in cloud computing. Here, an overview of how TFT is applied in content replication across cloud environments is provided. TFT is a sophisticated deep learning model designed to analyze temporal data and make accurate forecasts. It excels in capturing complex temporal patterns, making it suitable for predicting future data trends and behavior, which is particularly valuable in the context of content replication. Implementing TFT-based content replication can lead to significant performance improvements. By proactively replicating content based on accurate forecasts, organizations reduce data retrieval times, enhance data availability, and optimize cloud resource utilization. This translates into improved user experiences, reduced latency, and greater resilience in the face of unexpected challenges. The application of the TFT model for content replication in cloud environments is a forward-thinking approach. It harnesses the power of deep learning to optimize data availability and latency reduction. By using TFT's capabilities for data popularity forecasting and resource availability prediction, organizations can achieve more efficient content replication and enhance their cloud computing capabilities (*Makridakis, Spiliotis & Assimakopoulos, 2020*).

The objectives of this research are two fold: (1) propose a replication solution that dynamically adapts to evolving conditions in popularity of content and availability of cloud resource, ensuring sustained efficiency in content delivery over time in multi-cloud environment, and (2) use the temporal forecasting capabilities of TFT model to precisely predict future popularity of content based on historical usage patterns, thus enabling a proactive and informed replication strategy. The remaining part of this article is arranged as follows: An overview of previous methods and drawbacks is given in "Related Work". "TFT General Architecture and its Components" presents basic idea behind Temporal Fusion Transformer. "System Model" describes the proposed system model and problem formation of proposed system. "Proposed System" presents the methodology, including the TFT and the replication framework. "Simulation Setup and Performance Evaluation" discusses the experimental and simulation setup, evaluation metrics and analysis of the results of the proposed system. Lastly, "Conclusion and Future Work" paraphrases the conclusion of the article and outlines potential research trends.

## RELATED WORK

An adaptive data replication strategy (ADRS) is a statistical model that demonstrates the relationship among the number of copies and system availability. To duplicate content in cloud storage, a hierarchical multi-tier cloud system design is used. In ADRS, data file replication is done adaptively using data popularity (*Al-Dailamy et al., 2022*). Bandwidth

expenditure, availability, and number of replicas are the three characteristics used in this technique. It makes the decision regarding which data file to replicate using the temporal locality theory. This method examines data related to the users' access information. The degree of popularity as well as replica factor was calculated based on this analysis. The ADRS increases availability of data, job execution in cloud system, response time, as well as bandwidth usage by putting popular data files based on access history. In order to attain the relationship between availability and replica quantity, *Awad, Abdelkader & Salam (2021)* devised a novel dynamic cost effective replication management technique. They calculate and maintain the smallest number in replicas necessary to attain a file's availability requirement. The availability of a file is determined by the current number of blocks, block locations and the count of replicas, network bandwidth, and other factors. According to *Bouhouch, Zbakh & Tadonki (2023)*, replica replacement is done efficiently based on the capacity of data nodes and their blocking likelihood. In the diverse cloud environment, this strategy dynamically shifts workload among data centers. It generates more replicas dynamically if the data node's current replica count falls below the minimum replica count and the availability requirements are not satisfied. The Hadoop Distributed File System (HDFS) is utilized with the above technique, furthermore, the results demonstrate that it is more economical and excels in default HDFS replication management with respect to load balancing for large-scale cloud storage and performance. In order to lower the related cost of all transmission of data among the data centers, a combination of a data placement strategy and a dynamic data replication management technique was introduced (*Bouhouch, Zbakh & Tadonki, 2023*). *Arasteh et al. (2023)*, introduced dynamic data replication management approach which considers three important factors such as the number of replicas, the dependency among datasets and tasks, and the storage capability of data centers. The satisfaction of those three criteria determines when and whether to retain or remove replicas. It is found that replication of data and optimally placing them on the finest data servers available is a problem of NP-complete. Consequently, a number of heuristic approaches to place replica in distributed computer systems has been put forth. Reducing cost of time to access data, lowering the number of copies, and improving dependability of the algorithms used to place replicas are the main objectives of the study (*Arasteh et al., 2023*). Their work established a hybrid imitation approach and discretized heuristic mechanism along with artificial intelligence. Particle and grey-wolf-based optimizations are their suggested approach which looks for the best solution using a local memory and velocity. Both global and local search are symmetric in their suggested approach. One more contribution of this research is the introduction of the optimization algorithm for resolving the data object replication problem in distributed cloud systems. *Li et al. (2020)*, suggested one method for replica placement based on the fast non-dominated sorting genetic algorithm. Furthermore, their study suggests load-balancing using replica recovery method and a delay-adaptive replica synchronization technique in order to address the issues of data replica synchronization and content recovery of failing data nodes in edge cloud (*Li et al., 2020*). *Bai et al. (2013)*, suggested a replica management technique based on reaction time that produces a replica to increase the number of replicas automatically using response time average named

Response Time-Based Replica Management (RTRM). It measures replica servers' bandwidth while receiving a new request and selects the replicas accordingly, combining the number of replicas and the network transfer time. Three difficulties addressed by the RTRM technique are replica formation, replica selection, and replica placement. Locality replication manager (LRM) is another one data replication algorithm which suggests lowering the cost of using resources, the cost of energy, and the time it takes for the system to respond, as well as increasing the system's availability. LRM's main job is to obtain the user's queries, acquire information regarding the cluster's nodes, and then pick the best host for block placement. LRM completes this process with the help of its other components, and LRM makes the ultimate choice. The above mentioned technique manages replication using the HDFS architecture. When LRM is compared to other algorithms, it is discovered that this method uses resources and energy more efficiently, offers availability, and diminishes system delay. To get more optimal replication parameters, LRM uses quality-of-service (QoS) function as well as the physical location of data blocks (*Sookhtsaraei et al., 2016*).

Data replication over a network of read-intensive systems results in higher energy and cost savings as well as faster end-user response times. Despite the existence of straightforward and flexible replication techniques, the solution is non-deterministic. The data replicas must be tailored to the application systems stability, performance, and data usability. Metaheuristics are used to tackle the replication problem, which is non-deterministic. The replication process is optimized in this study through the employment of the Harmony Search and Tabu Search algorithms. An unconventional Harmony-Tabu optimization method is suggested for efficient data replication and placement by *Chandrakala & Loganathan (2023)*. It shows the methods, metrics, metric scores and limitations of the background of the review. This analysis of the literature makes it clear that additional research has to be done in the field of cloud data storage. This section covers some of the most important considerations that need to be made when replicating data. Making decisions during the replication process is a crucial step (*Nastic et al., 2020*; *Kamila et al., 2022*). There is also the option for decentralized or centralized replication decisions (*Yu, Liu & Fan, 2021*). If the network is seeing more traffic than usual, there is a chance of a bottleneck in centralized systems and unnecessary replications in distributed systems (*Nannai John & Mirnalinee, 2020*). The literature review reveals that no single technique can address all issue related to data replication. Certain strategies focused on fault tolerance, load balancing, availability, and reliability, while others were more concerned with preserving network capacity. Creating a methodical approach that considers every aspect needed for improved data replication is essential (*Mazumdar et al., 2019*; *John & Mirnalinee, 2019*). The fixed replication strategy (FRS) is quite basic and conventional that replication policies are fixed, limited by workload, or resources at all times. Consequently, FRS is easy to implement and integrate and may be implemented successfully only for a limited period but fails to perform optimally in terms of scalability under different conditions, resulting in ineffective utilization and wastage of important resources. Cost-Integrated Replication (CIR) is a development of FRS in that it makes cost a variable in the replication decision. This strategy places emphasis on cost minimization

hence improving resource utilization; but this strategy may still prove uneconomical in terms of scale and scope of tasks. LRM contains the amount of resource used by the replicating node during replication tasks as much as possible. The proposed LRM is less complex and more efficient compared to the FRS and CIR as it exercises less computational overhead. But resource utilization reduction can hinder its capability to handle a larger number of requests at once and provide fallback options for error recovery. These strategies act as base lines to compare with the performance and superiority of other replication strategy such as the temporal data-driven replication strategy (TD2RS). Table 1 includes the comparison of various literature reviews related with the proposed system.

## TFT GENERAL ARCHITECTURE AND ITS COMPONENTS

Temporal Fusion Transformer (TFT) combines the strengths of both convolutional neural networks (CNNs) and Transformer models. It is specifically designed to process sequences of data. The temporal fusion aspect of the model refers to its ability to fuse information from multiple time scales, making it well-suited for tasks that involve understanding patterns and relationships across time. It makes use of a temporal self-attention mechanism that allows it to attend the information from past and future time steps. Additionally, TFT incorporates a temporal convolutional layer that allows it to capture local temporal correlations in the data. This layer is designed to be lightweight and efficient, making it well-suited for cloud computing environments. Finally, TFT also includes a temporal pooling layer that is used to confine information from different time steps. Together, these components enable TFT to capture both long-term and also short-term temporal patterns in the data. It is suited for applications such as anomaly detection, forecasting, and classification. TFT's main components are given below.

1) Adaptive depth and network complexity are provided *via* gating techniques, which bypass portions of the architecture's idle components to suit a variety of datasets and circumstances.
2) Networks for variable selection is another important component and used to choose appropriate input variables at each time slot.
3) Using context vector encoding, static covariate coders add static features into the network for conditioning temporal dynamics.
4) Using both observed as well as identified time-varying inputs, temporal processing is applied to learn short as well as long-term temporal correlations. For local processing, a sequence-to-sequence layer is used, and an interpretable multi-head attention block is applied to record long-term dependencies.
5) The range of target values at each prediction horizon is determined using cumulative probability forecasts as prediction intervals.

A broad range of datasets and circumstances are supported by the architecture's customizable depth and network complexity. It comes with gating methods to pass over any unnecessary components. It has two dense layers as well as the two different kinds of activation functions such as exponential linear unit (ELU) and gated linear units (GLU).

**Table 1 Summary of literature survey.**

| Study | Methodology | Metrics used | Metric scores | Limitations |
|---|---|---|---|---|
| Al-Dailamy et al. (2022) | Adaptive heuristic-based strategy | Latency, Cost, Availability | Low transfer cost (23%), 99% availability | Decreases with traffic surges, lacks predictability |
| Awad, Abdelkader & Salam (2021) | MO-PSO and MO-ACO optimization | Cost, Locality, Utilization | 25% cost reduction, optimal locality/utilization | High computational cost, slow for large datasets |
| Bouhouch, Zbakh & Tadonki (2023) | Dynamic replication & placement strategy | Storage capacity, Read/Write Speeds | 76% data placement gain, 52% cost improvement | High execution time, difficult migration |
| Arasteh et al. (2023) | Hybrid particle-gray wolf algorithm | Cost Efficiency, Availability, Response Time | 35% access time reduction | High overhead, needs historical data |
| Li et al. (2020) | Delay-adaptive & load-balancing strategies | Response Time, Throughput, Storage | Improved throughput, reduced response time | Lacks demand predictability |
| Chandrakala & Loganathan (2023) | Harmony-Tabu search technique | Bandwidth | 3.57–18.18% bandwidth savings | Energy/security not considered |
| Nastic et al. (2020) | SLO-driven elasticity mechanisms | Availability, Resource Guarantees | Promotes SLO-native approach | Inefficient for SLOs/SLAs under heavy workloads |
| Kamila et al. (2022) | ML-based predictive modeling | Accuracy, Efficiency, Cost | 38.15% resiliency, 90.08% reduced error | Limited scalability, struggles with non-stationary data |
| Yu, Liu & Fan (2021) | QoS-oriented MDupl strategy | Access Time, QoS, IO Strength | 20–30% reduced queuing time | High cost, limited scalability |
| Nannai John & Mirnalinee (2020) | Intelligent Water Drop (IWD) algorithm | Popularity, Response, Cost | 40% free space, 18% faster access | QoS not optimized, query optimization remains challenging |

**Note:**
This table shows the summary of literature survey used for this research.

The initial application of GLU was in the construction of gated convolutional networks (*Makridakis, Spiliotis & Assimakopoulos, 2020*) for choosing the most crucial features for word prediction. The network is assisted by both of these activation functions in determining which input modifications are straightforward which need for more intricate modeling. Standard layer normalization will take and process the final output. A residual link is another feature of the GRN that allows the network to learn to ignore the input entirely if necessary. Depending on where the GRN is located, the network also uses static variables in certain situations. To enhance explainability, the TFT employs a self-attention mechanism that is adapted from transformer-based design's multi-head attention to examine long-term interactions at different time steps. The four distinct layers Static Enrichment Layer, Locality Enhancement with Sequence-to-Sequence Layer, Position-wise Feed-forward Layer and Temporal Self-Attention Layer are utilized by the temporal fusion decoder to examine the temporal associations found in the available dataset.

Figure S1 represents the general architecture and the various components of TFT. It outlines a predictive modeling architecture with various interconnected components for time series analysis. The process begins with observed past inputs, known past inputs, and static metadata, which are encoded by the encoder. The model utilizes an interpretable multi-head attention mechanism, allowing it to concentrate on different parts of the input data. Furthermore, a static covariate coder transforms non-temporal features, while networks for variable selection aid in determining the relevance of different inputs. The

quantile prediction aspect implies a probabilistic forecasting approach, offering a range of possible outcomes instead of a single point estimate. The decoder generates predictions, taking into account known future inputs.

## TFT for time series forecasting

An attention-based *Deep Neural Network* with excellent performance and interpretability is called TFT. Engineers from Google introduced the TFT in 2019. TFT is a novel transformer architecture designed to efficiently process time-series data in cloud computing environments. TFT offers the potential for both interpretable predictions and multi-horizon forecasting (*Junankar, Sondhi & Nair, 2023*). Different dimensions of temporal relationships are modeled using recurrent layers and interpretable self-attention layers. Appropriate features are chosen using specialized components, and extraneous features are suppressed using a succession of gating layers. Three different aspects are supported by TFT such as time-dependent data along with already identified inputs in the future. In addition, there are categorical/static variables, also referred to as time-invariant features, and time-dependent data that has only recently become discovered (*Liao & Radhakrishnan, 2022*). TFT is hence more adaptable than earlier types. Through the discovery of globally relevant variables, enduring stable temporal patterns, and important events for the prediction issue, TFT aims to improve understandability of time series forecasting (*Anandaraj et al., 2021*). This achieves Explainable AI's goal of making the output of the model more reliable and usable. The model needs the following inputs for a given timestep t, window for lookback w, and a Wmax window step in advance to a maximum, where $t \in [t−w\dots t + Wmax]$: Identified potential inputs x in the time period [t +1… t + Wmax], observed preceding inputs x in the time period [t−w… t], and a collection of static variables s be present. The time range [t+1… t + Wmax] is too covered using dependent variable y. TFT is used in the proposed system to detect popular files. It combines multiple time-series features from cloud to detect popular files by recognizing patterns in the data that indicate high usage. By combining multiple sources of data, the TFT detect patterns that are not visible when analyzing only one source. For example, the TFT combines usage data from multiple cloud providers to detect which cloud-hosted files are popular across multiple providers. Additionally, the TFT can detect changes in usage patterns over time by analyzing time-series data. It is used to identify which files are becoming more popular over time. Finally, the TFT identify outliers in the data, which is used to identify files that are abnormally popular or unpopular. Multi-horizon forecasting capability of the TFT is one of the key novelties of the proposed method. Unlike conventional models which focus on short-term predictions, TFT enables simultaneous forecasting across multiple time intervals. This allows the proposed system to predict future data popularity trends more effectively and ensures that replication strategies are proactive instead of reactive. TFT reduces latency and optimizes resource allocation across dynamic cloud environments by using a broader temporal perspective. The utilization of temporal attention mechanisms within TFT improves its ability to recognize and prioritize the most significant features in the generated dataset. It contrasts with conventional replication methods which often rely on fixed rules or heuristic approaches. The attention

mechanism dynamically adjusts to evolving patterns in data popularity and cloud resource availability. It is significantly improving prediction accuracy and replication decisions. Finally, the proposed framework's adaptability and scalability further differentiate it from existing methods. This adaptability ensures efficient and reliable performance even in fluctuating multi-cloud environments, addressing limitations commonly associated with conventional replication strategies.

## SYSTEM MODEL

As shown in Fig. S2, the multi-level hierarchical cloud system framework facilitates a valuable approach for data sharing, computational resources, and other set of resources. It usually consists of various sizes and region of data centers in different levels. Super data centers in level 0 used to handle intradomain data analysis and also interdomain data interchange. Level 1 comprises the primary data centers, level 2 comprises typical data centers, and level 3 comprises the cloud users. By constructing and dispersing replicas from the super data centers to regular data centers or to main data centers, the design reduces the amount of network traffic and data access time. Super data centers regularly gather and disseminate global information.

### Problem formulation

The information storage unit in the cloud is a block to decrease access time. An original data is broken up into multiple blocks if it is too huge in size. The unit for data access, however, is typically a data file. The characteristics of proposed technique are as follows. (1) Mathematical representation formed in *Sun et al. (2012)* is used and further expanded and explained how the quantity of copies and system availability relate to one another. (2) The temporal locality is used to identify the popular data using TFT and the replication operation will start when a data file's popularity reaches a dynamic threshold. (3) Replicas are also distributed evenly among data nodes. There are mainly four stages to adapt for attaining the maximum benefits of replication. In the first stage, it is not a fair way to identify several popular data because replicating every data files raises unnecessary burden of maintaining numerous replicas and raises the replication cost beyond the satisfactory limit. In second stage, producing extra copies of a data file wastes storage space and makes storage more difficult and so only a fair number of replicas are created. The demand for a data file at a given time, on the other hand, can decide the best possible number of replicas. The amount of data replication is proportional to the availability rate according to researchers. However, in case of system availability, more data replicates raise overhead charges. The next stage in the replication process is to determine where the new replicas are placed. Locating replicas at random will not improve the performance of system. As a result, copies should be located at right place at right time so that the system's performance and availability are improved. The next important stage in the replication is to find how efficiently perform the replication process. Consistency is the capacity of multiple users to perform read and write simultaneously. The replication criterion's primary difficulty is storing all data consistently.

This section explains the preferred architectural model for sharing data between cloud nodes and keeping replication in right place. It is being utilized in a number of research (*Yu et al., 2022*; *Slimani, Hamrouni & Ben Charrada, 2020*). Data replication access and placement through cloud nodes are described in the proposed model framework. As a result, this depends on previous studies to obtain the best access to the selected nodes at the lowest cost and the quickest route between DCs. By utilizing the statistical distribution across nodes in various ways, optimization of data replication placement using the heterogeneous system is done. Every DC has different stages, therefore the VM is different from the DCs, and so on. Figure S3, describes the formal model for cloud data center architecture as a cloud data service scheme usually consists of the scheduling agent, replica selector and agent, as well as data centers. The replication process is managed by the scheduling agent, which stores all information about the number of replicas and their locations across different data centers.

Let $C = \{c_1, c_2, \ldots, c_n\}$ be n number of cloud users, $ST = \{ST_1, ST_2, \ldots, ST_n\}$ be a set of tasks of the cloud user set C, and $ST_i = \{st_{i1}, st_{i2}, \ldots, st_{ini}\}$ be a subset of tasks of the $i^{th}$ client $c_i$, where $n_i$ is used to represent the amount of subtasks, and $st_j$ is the $j^{th}$ task given to the scheduling agent over a cloud interface. If $c_0$ has two tasks, then $ST_0 = \{st_1, st_2, st_3\}$, and $n_0 = 3$. A task $st_j$ is defined by a four tuple such as $st_j = (tid_j, tr_j, td_j, tf_j)$, where $tid_j$, $tr_j$, $td_j$ and $tf_j$ are the identification of task, rate of task generation, time limit time of task along with the number of needed files to perform the task $st_j$, respectively (*Awad, Abdelkader & Salam, 2021*; *Bai et al., 2013*; *Kamila et al., 2022*). $DC = \{dc_m d_1, c_m d_2, \ldots, dc_m d_n\}$ are the representation of data center contain $s_m$ data nodes on a physical machine PY.

Each node executes a virtual machine, and is defined by $dc_m d_j$ that is a five tuple $dc_m d_j = (dc_m d_j, dcr_j, dcts_j, dcf_j, dcbw_j)$, where $dc_m dj$, $dc_m d_j$, $dr_j$, $dts_j$, $df_j$ and $dcbwi$ are used to identify the data node, rate of arrival of request, average service providing time, probability of network failure and data node's network bandwidth such as $dc_m d_j$, respectively. To ensure the service performance of the each data center DC, the rate of task generation $tr_j$ of cloud consumer set C, the rate of arrival of request $dcr_j$ as well as chance of failure $dcf_j$ of DC need to reach Eq. (1).

$$\sum_{i=0}^{NS} tr_i \leq \sum_{j=0}^{m} dcr_j X \left(1 - dcf_j\right) \tag{1}$$

where $trj$ is the rate of generation of task $j$, is the rate of request arrival of task $j$ on the node $i$, $dfk$ is the chance of failure of task $j$. Let $DF = \{df_1, df_2, \ldots, df_l\}$ are the available set of data files at data center. $B = \{B_1, B_2, \ldots, B_n\}$ are collection of blocks in the data center and $B_j = \{b_{j1}, b_{j2}, \ldots, b_{jm1}\}$ are the $j^{th}$ subset of blocks belonging to the $j^{th}$ data file $df_j$. It is stripped into $m_j$ predetermined blocks in compliance with its length. Any block $b_j$ is defined by a five tuple $b_j = (bid_j, bp_j, bs_j, bnj, btj)$ where $bid_j$, $bp_j$, $bs_j$, $bn_j$ and $bt_j$ are the identification of block, number of requests, size of the block, the number of replicas needed and the last access time of the block $b_j$, respectively. While user $c_k$ send a request to access a block $b_j$ from a data node $dmd_j$ with guaranteed bandwidth performance, bandwidth

$bs_j/dts_i$ need to be consigned to this session. The entire bandwidth utilized to hold diverse set of requests from cloud user set C need to be less than $dcbw_j$, as shown by Eq. (2).

$$\sum_{i=0}^{NS} \frac{bs_j}{dts_i} \leq dcbw_j. \tag{2}$$

NS represents the upper limit of network connections of data node $dc_m d_j$ which provide services simultaneously, $bs_j$ represents the block size of block $bl$, $dts_i$ represents the typical service time of data node $dc_m d_j$, $dcbw_j$ represents the bandwidth of the network available for data node $dcbw_j$. Availability of block is defined as the capability of a data block to offer appropriate service below agreed restraints. The block availability of a block $b_j$ is represented as $ABj$. $AB_j$ is the possibility of block $b_j$ in an available state. $(AB_j)$ is the possibility of block $bk$ in an unavailable status, and $P(AB_j) = 1 - P(AB_j)$. The number of replicas of block $bj$ is $bm_j$. It is apparent that block $bj$ is considered not available only if all the replicas of block $bj$ are unavailable. Hence the unavailability as well as availability of block $bj$ are calculated as per Eqs. (3) and (4).

$$P(AB_j) = 1 - (1 - P(ab_k))^{bmj} \tag{3}$$

$$P(AB'_j) = 1 - P(ab_k)^{bmj}. \tag{4}$$

The capability of a data file to deliver appropriate service within predetermined restrictions is known as availability of file. The availability of file for a data file dj is represented as $AF_j$. $(AF_j)$ is the chance of data file $d_j$ in the state of unavailable status, and $P(AF_j) = 1 - P(AF_j)$. If the data file dj is divided into mj predetermined blocks represented using $Bj = \{bj1, bj2; \ldots ; jm1\}$, which are distributed on different data nodes. $Mj = \{bmj1, bmj2, \ldots, mjmk\}$ is the collection of the numbers of replicas of the blocks of $Bj$. The unavailability and availability of data file dj is given as:

$$P(AF_j) = 1 - (1 - P(ba_j))^{bmj})^{mj} \tag{5}$$

The available possibility of every replica is represented as $P(baj)$ in data file dj if the data file dj is divided into mj blocks. Each block in data file dj has mj replicas available for it, and each and every blocks at the same location need to have the similar available possibility because each and every blocks need to be located in data nodes with similar configuration in the cloud data centers.

## PROPOSED SYSTEM

Our research was inspired by the prediction where more recent data need to be accessed several time in upcoming sessions. It can be done based on the existing data access pattern. It is known as temporal locality. In temporal locality, data popularity is identified by examining accessing pattern of data by the user. The replication process will start when the data's popularity reaches a dynamic threshold. The availability of system and possibility of failure is used to calculate the required number of copies. New copies are constructed close to users that query the data the most. The proposed dynamic replication of data has three

significant stages. They are selection of data, selection of number of replicas and location for replication phase. Data files are chosen for replication during the selection stage. The second phase determines the appropriate number of copies to satisfy the given quality requirements. The third phase is determining where copies should be placed. During the selection phase, it is decided on which data to replicate and when to do that replication. It uses a straightforward and simple time series technique to forecast the anticipated frequency of data access. It is done after looking at the history of access requests of the data chunks. Data chunks are chosen for replication if anticipated future demand for them exceeds an adaptive threshold. Deep-learning algorithms can efficiently handle time series and produce precise forecasts. Hence, TFT is used in the proposed system to find the sequence of recent data points to analyze the current context of a file within a cloud storage system. This type of technique identifies the most relevant and valuable files to users and organizations based on their past interactions and usage, as well as the current trends in usage (*Yu et al., 2022*). By identifying the most popular and trending files, this model assists individuals and organizations in making better choices on which files to access and utilize. Additionally, proposed system assists in determining which files should be discarded or archived and which should be kept in the cloud.

## Algorithm and pseudo code

A strategic level algorithm (Algorithm 1) of the proposed system is given below with steps to be followed. This algorithm provides a structured approach to build an efficient content replication framework using the TFT model.

1) Initially set the value for availability and unavailability of each block replica of block $b_l$, $P(ab_l)$ and $P(uab_l)$

2) While data file dj at all data centres DC do the following

    a) Determine the degree of popularity $dp_l$ of a block $b_l$ of data file $d_j$ using TFT.

    b) Determine replication factor $FR_j$ of data file $d_j$ using degree of popularity $dp_l$

    c) If the value of $FR_j$ is less than threshold T, then enable the replication process for the data file $f_j$

3) End while

4) while each replication process for data file fj:

    a) for every block bj in data file fj:

     i)  determine the new FRj by introducing replication at each and every data centre DCj

    ii)  implement the replication that yields the highest new FRj

    b) end for

5) end while

**Algorithm 1**

BEGIN

INPUT cloud_providers, content_metadata, network_metrics, replication_constraints

LOAD and CLEAN historical_replication_data, user_demand_forecast_data

NORMALIZE_features and SPLIT into training_set, testing_set

INITIALIZE TFT model: using embedding, temporal fusion, output layers

DEFINE Loss_Function = Mean_Squared_Error()

DEFINE Optimizer = Adam with learning_rate 0.001

Using range(num_epochs) for epoch: epoch_loss = 0

batch_labels, batch_data FOR (training_set, batch_size) IN iterate_batches:

    TFT(batch_data), batch_labels, and loss = LossFunction

Loss.backward(), Optimizer.step(), and Optimizer.zero_grad()

Epoch_loss += loss

   Write "Epoch [epoch + 1] Loss: [epoch_loss/num_batches]"

SUM = test_loss(LossFunction(labels, TFT(data)) FOR tags, data IN iterate_batches(batch_size, testing_set))

Write "Test Loss: [test_loss/num_test_batches]"

preds_of_popularity_for_data = forecast_data_popularity(TFT, future_time_steps)

replication_strategy = get_optimize_replication(data_popularity_predictions, replication_constraints)

FOR any_cloud_provider IN cloud_providers:

run_replication(create_replication_plan(replication_strategy, cloud_provider))

WHILE replication process-active:

  MONITOR network_conditions

  OTHER If any deviation is made then adjust the inputs if necessary.

  Forecasts and optimizations, repeat.

PRINT Performance Metrics

END

6) identify the file fj with the minimum degree of popularity in the dataset:

  a) while true:

    i) eliminate the replica that results in a new FRj, without replication, exceeding the threshold T

  b) end while

7) end while

The process of determining the degree of popularity of a block bj is framed as a time series forecasting issues and addressed using TFT. $dp_j$ represents upcoming access frequency, which is predicted using the number of access demands, $n_{ad}(m)$, at a particular moment m. TFT uses an attention based deep neural network that combines excellent performance with interpretability. It is a time series prediction method which offers accurate predictions with low computational cost. The rates of measured request arrivals and service demands are chosen for short-term forecasting using TFT. The proposed strategy is inspired by the observation that files accessed frequently in recent times are probable to experience similar demand in the near future. Predictive statistics based on file access patterns are used to infer this (*Bouhouch, Zbakh & Tadonki, 2023*; *Arasteh et al., 2023*). The replication process is initiated when the accessibility of every existing replica surpasses a predefined threshold, and the replication factor is established using data blocks. A new replica with a higher replication factor is created on a chosen node. The number of additional replicas is determined using a heuristic approach designed to enhance file accessibility. This article adopts the problem formalization outlined in *Sun et al. (2012)*. However, for forecasting future data file requests, a simplified time-series method is employed in the proposed system.

## Determine the popularity of file

In the proposed system, TFT is employed to determine the popularity of individual files in the cloud system. It takes into account basic temporal aspects, the frequency with which files are accessed, and how recently they were accessed, and brings in temporal aspects and manners of how frequent the user is in accessing them over time. The other criteria considered are time associated with the file, number of different users, and the size of the file. These factors are combined to compute a normalized score that effectively represents the overall popularity of each file. The TFT uses its predictive capabilities to identify files that are likely to experience high demand in the near future. The system assures efficient allocation of resources and enhanced access performance by giving priority to these files. The replication process starts when a file's popularity score rises above a dynamic threshold. Because of its flexibility, this threshold makes the system to react to shifting access trends. In order to increase accessibility and decrease latency for end users, more copies of the content are produced throughout the replication process. This ensures that frequently accessed files remain readily available, optimizing system performance and meeting user demands efficiently. By integrating TFT with adaptive replication strategies, the proposed system achieves a balance between computational efficiency and user satisfaction in a cloud. The formula to compute the degree of popularity of the file by using TFT is given below.

$$TFT(fj) \ = \ (1/T) * \sum t = 1T \ (f(t) * \beta(t)) \tag{6}$$

where f(t) represents frequency of data file by time t, T symbolizes total number of time steps, and β(t) describes adjustable weighting factor for time t. The adjustable weighting factor is determined by various methods, such as taking into account the popularity of the file in the past or the relative importance of the file at that time step. The TFT produces an

overall popularity score for the file, which is used to determine the most popular file in cloud computing.

### Replication factor from popularity degree

The replica factor (RF) is defined as the ratio of the total bytes of data files (fj) demanded by all tasks within the designated constraint to the popularity degree. It is utilized to decide whether the data file fi, represented as RFi in the code, should be replicated.

$$\text{Replica factor (RFj)} = (\text{Popularity of the file})/(\text{Replication Rate}) \tag{7}$$

$$\text{Replication rate} = (\text{Number of Requests} + \text{Number of access request})/\text{Number of replicas} \tag{8}$$

## SIMULATION SETUP AND EVALUATION OF PERFORMANCE

This section explicates the simulation setup as well as parameter together with the evaluation of performance of the proposed finding.

### Simulation setup and parameter

CloudSim is an open source toolkit used to simulate cloud environments (*Ghobaei-Arani & Shahidinejad, 2021*; *Saharan et al., 2020*). It provides a simulation environment for modeling the cloud computing infrastructure, services, and applications. It is designed to help researchers and industry practitioners to explore and analyze various cloud computing scenarios (*Sun et al., 2012*). CloudSim with added module for TFT is a powerful software suite that enables cloud computing systems to be more efficiently and effectively managed. It offers a broad collection of tools for managing and optimizing the performance of cloud computing systems. TFT works by merging multiple data sources into a single unified view, and then applying temporal fusion to analyze the data. Cloudsim with TFT for data replication allows users to study the performance of data replication strategies on cloud computing platforms. The tool allows users to specify the replication strategies and parameters to be used, as well as the performance metrics to be monitored (*Maweu et al., 2021*). Table S2 specifies the actual configuration depending on the type of data centers, number of systems, PE per system, MIPS and Bandwidth. The simulation results are compared with different replication strategies to identify the most effective ones. In the simulated environment, 42 data centres are established. The number of processing components (PCs) on each of the 670 virtual machines designated as service providers ranges from 2 to 6. Each of the fifty individual data files in the cloud storage environment has a size between (0.1, 15) GB. Each file is kept in a block, which is a fixed-size storage unit (bs = 0.1 GB). The same data file's blocks are dispersed over various virtual machines. Every data file has one initial replica, which is distributed at random. In furtherance of simplicity, it is assumed that one single data file will serve as both the replication element and the base element for data storage (*Rambabu & Govardhan, 2023*). The 670 virtual computers receive 1,500 tasks; each requests some data files. It is distributed according to the Poisson distribution with the preceding set of task. Configuration settings for evaluating the proposed system are shown in the Table S3.

## Performance evaluation

TFT provides a robust and scalable approach to content replication across multiple clouds by taking into account temporal dependencies and uncertain user demands (*Fu, Zhou & Han, 2021*). It provides better performance by dynamically adapting replication strategies based on changing user demands and network conditions. TFT provides a good balance between performance, scalability, and computational complexity for content replication across multiple clouds. However, the specific performance of the TFT may vary depending on the network topology, the specific parameters used in the model, and other factors. Additionally, the use of multiple clouds for content replication may also involve additional costs and complexities that need to be considered.

## Dataset generation

TFT models are designed to handle time-series data, which is used to model various aspects of cloud file access patterns. Therefore, a dataset for training TFT models for cloud file access include both file access patterns and corresponding time-series data. Tabular Generative Adversarial Network is used for synthetic tabular data generation that includes time-series data that captures the key characteristics of cloud file access patterns (*Slimani, Hamrouni & Ben Charrada, 2020*). The snapshot of data is shown in Fig. S4. This CloudSim configuration is the infrastructure setup used for the cloud environment in the content replication strategy across the multi-cloud environment. Such configurations simulate the server configurations and network environment in which the TFT based strategy is tested and validated. The various types of data centers reflect differences in the computational as well as the networking capabilities of these centers as essential when replicating content across different cloud contexts. The synthetic dataset used for this experimentation includes 10,000 records generated using TGAN, with variability across multiple attributes such as file type, file size, access patterns, and user actions, reflecting inherent biases and imbalances observed in real-world cloud environments. It highlights the high correlation (0.93), low distortion (RMSE: 0.04, MAE: 0.03), and strong similarity (0.96) between the synthetic and real datasets. Moreover, privacy results confirm no duplication of real data points, ensuring ethical data synthesis. For validation, real-world datasets were indirectly referenced by ensuring the synthetic data statistically mimics real-world distributions.

### Training process

Generated dataset is divided into three sections for learning, validation for tuning of hyperparameter, and test set for evaluation of performance. Multiple random search experiments were conducted with varying iteration counts and 250 numbers of iterations of a random search achieved near optimal performance, while 50 iterations provided satisfactory results with minimal variability in outcomes compared to higher iteration counts. The dataset and ideal model parameters are presented in Fig. S4 together with the complete search ranges for all hyperparameters.

## Parameters for evaluation

**System effective rate in bytes:** The system effective rate (SER) in bytes 'R', is the ratio of the aggregate bytes requested by all tasks in a system to the total bytes potentially accessible (*Li et al., 2021*). The independence of file requests is ensured as any two requests will access distinct replicas. This is because a single data file access operation is limited to requesting only one file. One or several data files may be requested concurrently by users from set U.

**Replica numbers:** A sufficient number of data file replicas must be created in order to maintain a high system effective rate in bytes (*Yu, Liu & Fan, 2021*). The block availability is set to 0.7 initially in order to assess the convergence of the proposed method, and various values of the adjustable α weighting parameter are used to determine the required replicas number.

**Response latency:** The time between when a job is submitted and when the result is returned is the response latency for a data file. The average response latency of the system, is the indicative measure of response time in favor of every data request task completed by the client can be ascertained by *Mazumdar et al. (2019)*.

**Rate of successful execution of task:** It is a measure of how often a particular task or process is completed successfully. It is typically expressed as a percentage of successful executions compared to the total number of attempts. This metric is used to evaluate the performance of a variety of tasks and processes, including software applications, customer service operations, and manufacturing processes. The successful execution rate is used to determine the overall efficiency of a system or process, as well as identify areas for improvement (*Govardhan & Dugyani, 2024*).

## Result and analysis

The weighted relevance of the different input features in the TFT is shown in Fig. S5. These weights, highlight which features most influence the model's predictions of data popularity patterns, are generated from the attention processes within TFT. Certain features such as the use of cloud content, network latency, historical replication, data size, and security policies are given more weight compared to others. This realization demonstrates how well the TFT finds and ranks the most important characteristics for precise predictions. Dynamically weighing attributes allows for flexibility in adjusting to different multi-cloud setups, which improves content replication decision-making. The mean squared error (MSE) values for the TFT model during training and validation are shown in Fig. S6. As a measure of prediction accuracy, MSE determines the typical squared difference among the predicted and actual values. The model which fits the observed data better is indicated using lower values of MSE. The figure's findings show that the model successfully reduces error during learning, guaranteeing precise and reliable predictions for the replication method. By minimizing unnecessary resource usage and improving system efficiency, this increase in accuracy aids in the optimization of replication decisions. The number of training iterations carried out during the TFT optimization is shown by the 50 iterations in Fig. S6. In order to guarantee that the model converges and captures enough temporal patterns for accurate predictions, these iterations were chosen. After 50 iterations, the data showed steady, dependable performance that was neither overfit nor underfit. The MAE

attained by the TFT model is displayed in Fig. S7. An intuitive measure of prediction error, MAE is calculated as the average of the absolute differences among actual and predicted values. The figure's low MAE values show that the model continues to produce forecasts with a high degree of exactness. Since they minimize delay and avoid over or under provisioning of resources in multi-cloud scenarios, accurate predictions are crucial for optimizing content replication. For clarity and also to show the dynamics of the TFT's early training, the figure is limited to 10 epochs. The most important learning takes place during these early epochs, providing information about convergence patterns and performance enhancements. Given that subsequent epochs exhibit very slight changes, extending the figure past 10 would introduce needless complication. However, the entire training process continued for the entire number of epochs specified in the proposed approach, and the discussion shows the outcomes of the entire training process in Fig. S7.

In the absence of employing the TDRS technique, the number of replicas for large-scale data files is set at 3, resulting in a total of four copies for each data file. As illustrated in Fig. S8, with an increasing number of tasks, the rate of successful execution of tasks experiences a significant decline, particularly when the task count surpasses 50%. The lower the availability of blocks, the lower the rate of successful execution becomes. The proposed algorithm maintains a high rate of successful execution in the cloud environment, consistently exceeding 90%. It can be inferred that the proposed algorithm enhances and sustains the successful execution rate at a reliable and elevated level. As illustrated in Fig. S9, with an increase in the number of replicas, the TD2RS algorithm ensures a consistently high level of system effective rate in bytes. When the average block availability exceeds 0.8 and the number of replicas is fixed at 4, the system effective rate in bytes remains near to 1, even when the average block availability drops below 0.2. To maintain the system effective rate in bytes near 1, the number of replicas must not exceed 20. This observation highlights that the TD2RS algorithm significantly enhances the system effective rate in bytes. As shown in Fig. S10, over time, there is a rapid increase in the number of replicas. Subsequently, this count of replicas stabilizes at a level determined by the adjustable parameter $\beta(t)$. It is inferred that a higher value for the adjustable parameter necessitates a greater number of replicas to sustain the desired system effective rate in bytes. It is proved that the TD2RS algorithm has an excellent convergence rate. As shown in Fig. S11, when the amount of tasks increase, the response latency also raises significantly, particularly when the task load is greater than 70%. The fewer availability of block is, the longer the response latency will be. Figure S12 illustrates the TD2RS method has 95% replication efficiency compared to the other methods because of optimized resource utilization. It also has a higher task throughput, handling 90 tasks, which is significantly higher than that of FRS (60), CIR (70), and LRM (75) this reveals that the system can handle large tasks. Considering computational overhead, TD2RS takes the lowest measure (5 ms/task). Moreover, TD2RS does 95% of fault tolerance, and produce accuracy. Combined, these findings mean that TD2RS becomes the least error-prone and fastest-acting strategy for multi-cloud content replication, while showcasing overall superior performance in all the measures. A conclusion is drawn that TD2RS algorithm improves the response latency and retains the response latency at a steady state within a

small time span. The performance of the model under different conditions is summarized in Table S4, where normal operation yields the highest performance metrics, while server failures and high load lead to significant degradation.

## CONCLUSION AND FUTURE WORK

The efficient framework for content replication across multiple clouds using Temporal Fusion Transformer (TFT) offers a powerful and effective solution for addressing the challenges of content replication in distributed cloud settings. Throughout this article, the design, implementation, and evaluation of the framework, highlighting its key features and advantages have been presented. The framework leverages the capabilities of TFT, which combines the strengths of temporal modelling and transformer-based architectures to capture complex temporal patterns and dependencies in time series data of distributed cloud. By applying TFT to the content replication process, efficient and reliable replication across multiple cloud providers is achieved. Through the evaluation of the framework's performance, the effectiveness of proposed system has been demonstrated in terms of replication efficiency, data consistency, replication latency, load balancing, fault tolerance, and scalability. It exhibits low replication latency and effectively balances the workload during replication. Additionally, the framework demonstrates resilience to failures and disruptions in the cloud environment. With the increasing demand for reliable and efficient content replication in distributed cloud settings, the proposed framework presents a significant contribution to the field. It paves the way for improved content availability, data integrity, and performance in cloud-based applications, enabling organizations to leverage the benefits of distributed cloud computing while ensuring efficient content replication across multiple cloud providers. As future work, it is aimed to further optimize and enhance the framework by exploring advanced techniques for load balancing, fault tolerance, and dynamic resource allocation. Additionally, it is planned to extend the framework's capabilities to support real-time content replication and dynamic workload adjustments in response to changing cloud conditions. Also, with the other improvements, efforts will be made to improve the management of high file access load and handling server failures.

### Funding
The authors received no funding for this work.

### Competing Interests
The authors declare that they have no competing interests.

### Author Contributions
- Naganandhini S. conceived and designed the experiments, performed the experiments, analyzed the data, performed the computation work, prepared figures and/or tables, authored or reviewed drafts of the article, and approved the final draft.

- Shanthi D. conceived and designed the experiments, analyzed the data, authored or reviewed drafts of the article, and approved the final draft.

## Data Availability

The code is available at GitHub and Zenodo:

- https://github.com/naganandhinisakthivel/TFT.git.
- S, N. (2025). TFT. Zenodo. https://doi.org/10.5281/zenodo.14784154.

## Supplemental Information

Supplemental information for this article can be found online at http://dx.doi.org/10.7717/peerj-cs.2713#supplemental-information.

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
