# Peer review of "Temporal fusion transformer-based strategy for efficient multi-cloud content replication"

_PeerJ Computer Science, doi:10.7717/peerj-cs.2713_

## Round 0.1 · original submission · Major Revisions

· Academic Editor

Major Revisions

Please also include a summary of each comment and action taken by you for all actionable suggestions made by the reviewer.

Reviewer 1 ·

Basic reporting

A comparison table for literature review section would improve the clarity of the novelty of solution. A table showing the methods, metrics, metric scores and limitations of the background studies might be added.

Avoid using long sentences. Instead use shorter sentences and present more precise narrative. Especially the last paragraph of "Proposed System" section is hard to grasp and needs revision.

The proposed pseudocode, starting at line 405, needs revision. The current state of the pseudocode looks more like a Python code than a pseudocode.

The sentence in line 483 "The TFT algorithm can be used to 484 identify popular files and prioritize them accordingly" is not clear whether this algorithm is the proposed algorithm or not.

In the same sentence, "can be used" makes the sentence unclear whether the TFT is utilized in the solution or not. Please do not use words like "can be" while talking about proposed study. There are more occurences, revise all of them accordingly.

The sentence in line 485 "The replication operation will start when a 485 data file's popularity reaches a dynamic threshold." is written using future tense. Instead, you may use simple present tense.

Experimental design

no comment

Validity of the findings

Give a proper for interpratations for figures 5, 6, 7. Which models surpass others and what does it mean? What was the hypothesis and does these results support that?

In line 559, what are the 50 iterations are for? What are others, state clearly.

Figure 7, why is figure limited to 10 epochs?

Increase the resolution and quality of Images, they are hard to read.

Additional comments

The authors suggest a TFT-based solution for data replication problem in cloud computing. The TFT method is quite novel for the problem, combining forecasting and replication. The paper is overall well structured. The literature review presents sufficient background references. The use of TFT model for data replication is a novel approach. The study is within aims and scope of the journal. The reporting is acceptable. Still, there are some improvements that can be made regarding both the narrative and presentation.

Reviewer 2 ·

Basic reporting

The manuscript is generally well-written, but some areas require improvements in terms of clarity and grammatical accuracy. The abstract, while informative, should explicitly state the research outcomes to highlight the key findings. Additionally, the introduction could be enhanced by incorporating recent advancements in content replication strategies for dynamic environments. Simplifying technical terms and ensuring consistent use of terminology throughout the manuscript would improve readability.

- The abstract should include specific research outcomes to clearly highlight the key findings.
- The introduction requires a more detailed discussion of recent advancements in content replication strategies to strengthen the literature review.
- Simplify complex sentences and ensure consistent use of terms like "TFT model" and "replication strategy."

Experimental design

The methodology is clearly described, but the novelty of the approach should be emphasized more strongly, particularly in terms of how the proposed framework surpasses traditional methods. While the use of synthetic data is innovative, the dataset should be described in greater detail, and real-world datasets should be considered to validate the findings. Additionally, the system’s adaptability to changing workloads and resource availability should be elaborated upon to explain how quickly and frequently the system reacts to real-time conditions.

- The novelty of the methodology should be emphasized, particularly focusing on how TFT’s multi-horizon forecasting outperforms existing methods.
- More details about the synthetic dataset size and variability are needed, and real-world datasets should be considered for validation.
- The adaptability of the system to real-time changes and extreme conditions should be explained more thoroughly.

Validity of the findings

The results are promising but would be strengthened by more detailed comparisons with recent methods, such as CIR and LRM . The comparison lacks depth in terms of performance metrics like replication efficiency, scalability, and computational costs. Including comparative tables or figures would enhance the argument. Additionally, the paper should provide more information about how the system handles extreme or unpredictable events, as this is important for demonstrating the framework's robustness and fault tolerance.

- More detailed quantitative comparisons with recent methods like CIR and LRM are needed to strengthen the argument.
- Comparative tables or figures showing differences in replication efficiency, scalability, and computational costs would improve the analysis.
- The system’s fault tolerance and ability to handle extreme conditions should be better explained.

Additional comments

None

---

## Round 0.2 · Minor Revisions

· Academic Editor

Minor Revisions

After I review the changes you make, I will make the final decision. Thanks for your interest in the journal.

Reviewer 1 ·

Basic reporting

There are some writing issues such as (not limited to),
- You need to leave a space after a comma: computing,particularly
- You need to leave a space when starting a new sentence: strategy.Remaining
- There are words where no space is left between: organizationsreduce
The pseudocode given between lines 434-487 is not a pseudocode. Especially after step 3. Please revise these errors.

Literature review tables contain summary of information, which provide overall view of the state-of-art. You may revise cell contents with keywords (to summarize the information and make reading it easier). The detailed explanation can be given in the text.

Experimental design

no comment

Validity of the findings

no comment

Reviewer 2 ·

Basic reporting

The revised manuscript significantly improves on the points raised in the first review report.

Experimental design

The authors have improved the accessibility and clarity of their study by emphasising the use of TFT for multi-horizon prediction and revising the pseudocode to reflect high-level steps. They have also included tables and figures comparing TD2RS with CIR and LRM, addressing the issue of partial validation.

Validity of the findings

The authors describe a synthetic dataset and justify its use due to its similarity to real-world scenarios. They use figures and tables to compare the proposed method (TD2RS) with existing approaches. They include metrics like latency reduction and data availability improvement.

Additional comments

None

---

## Round 0.3 · accepted · Accept

· Academic Editor

Accept

Thanks for your prompt attention to this matter and making the required changes.